# Characterization of a Virally Encoded Flavodoxin That Can Drive Bacterial Cytochrome P450 Monooxygenase Activity

**DOI:** 10.3390/biom12081107

**Published:** 2022-08-11

**Authors:** David C. Lamb, Jared V. Goldstone, Bin Zhao, Li Lei, Jonathan G. L. Mullins, Michael J. Allen, Steven L. Kelly, John J. Stegeman

**Affiliations:** 1Faculty of Medicine, Health and Life Sciences, Swansea University, Swansea SA2 8PP, UK; 2Biology Department, Woods Hole Oceanographic Institution, Woods Hole, MA 02543-1050, USA; 3Cleveland Clinic Lerner Research Institute, 9500 Euclid Avenue, NB21, Cleveland, OH 44195, USA; 4Department of Biochemistry, Vanderbilt University Medical School, Vanderbilt University, Nashville, TN 37232-0146, USA; 5Department of Biosciences, College of Life and Environmental Sciences, University of Exeter, Stocker Road, Exeter EX4 4QD, UK

**Keywords:** flavodoxin, virus/phage, cytochrome P450, evolution, *Bacteria*

## Abstract

Flavodoxins are small electron transport proteins that are involved in a myriad of photosynthetic and non-photosynthetic metabolic pathways in Bacteria (including cyanobacteria), Archaea and some algae. The sequenced genome of 0305φ8-36, a large bacteriophage that infects the soil bacterium *Bacillus thuringiensis*, was predicted to encode a putative flavodoxin redox protein. Here we confirm that 0305φ8-36 phage encodes a FMN-containing flavodoxin polypeptide and we report the expression, purification and enzymatic characterization of the recombinant protein. Purified 0305φ8-36 flavodoxin has near-identical spectral properties to control, purified *Escherichia coli* flavodoxin. Using in vitro assays we show that 0305φ8-36 flavodoxin can be reconstituted with *E. coli* flavodoxin reductase and support regio- and stereospecific cytochrome P450 CYP170A1 allyl-oxidation of epi-isozizaene to the sesquiterpene antibiotic product albaflavenone, found in the soil bacterium *Streptomyces coelicolor*. In vivo, 0305φ8-36 flavodoxin is predicted to mediate the 2-electron reduction of the β subunit of phage-encoded ribonucleotide reductase to catalyse the conversion of ribonucleotides to deoxyribonucleotides during viral replication. Our results demonstrate that this phage flavodoxin has the potential to manipulate and drive bacterial P450 cellular metabolism, which may affect both the host biological fitness and the communal microbiome. Such a scenario may also be applicable in other viral-host symbiotic/parasitic relationships.

## 1. Importance

Herein, we demonstrate that virally encoded flavodoxin has the potential to manipulate and drive host bacterial P450 cellular metabolism affecting both host biological fitness and the communal microbiome. Viral phenotypic impact was thought limited to merely affecting host mortality. Recent work has cast this aspersion aside, as we have come to realize that close metabolic integration appears to be required for viruses to subdue and coerce their unwitting hosts. Our work herein clearly demonstrates the potential for viruses to not just manipulate ‘virus centric’ replication activities, but also broader host metabolic activities and cellular function(s) as well.

## 2. Introduction

Flavodoxins (Fld) are small water-soluble electron transfer proteins that contain non-covalently bound flavin mononucleotide (FMN) as the active redox component [1]. Flds were first discovered in the 1960s in both cyanobacteria and *Clostridia* growing in low iron conditions, where they serve as electron carriers, replacing iron-containing ferredoxin (Fd) in metabolic reactions leading to NADP+ and N_2_ reduction [2,3]. Since that time, Flds have been isolated from numerous *Bacteria*, *Archaea*, cyanobacteria and some eukaryotic algae [1]. Flds are located in the bacterial cytosol and algal chloroplasts and function in electron-consuming metabolic pathways and act as electronic switches between cellular sources of reducing power (e.g., light-driven reactions, pyridine nucleotides, sugars). Biochemically, all characterised flavodoxin proteins to date are highly acidic proteins consisting of ~140–180 amino acid residues (MWs~15–22 kDa) and structurally consist of a five-stranded parallel beta sheet enclosed by alpha helices on either side [1]. Catalytically, the Fld FMN cofactor is responsible for electron transfer, cycling between a neutral semiquinone (sq) and hydroquinone (hq) forms [4,5].

In bacteria the Fld gene is inducible, functioning as an adaptive resource under environmental or nutritional hardships (e.g., iron limitation). Furthermore, Fld has been shown to be an essential gene in some bacteria such as *Escherichia coli* [6] and *Helicobacter pylori* [7]. Although not present in higher eukaryotes such as plants and animals, a descendent of the Fld gene appears fused in some eukaryotic genes that encode multidomain redox proteins, e.g., cytochrome P450 reductase (POR) [8]. Flds are divided into short-chain and long-chain types differing by a 20-residue loop of unknown function. Based on phylogenetic analysis it has been proposed that the long-chain Flds may have preceded the shorter ones in evolution [9]. Long chain Flds are found exclusively in cyanobacteria and algae [10,11]. 

Genes thought to encode flavodoxin have been tentatively annotated in some viral (bacteriophage) genomes. In such phages, the flavodoxin gene is most often associated with genes encoding α and β-ribonucleotide reductase (RNR), and the gene products are thought to form a reversible protein complex that is responsible for converting ribonucleotides to deoxyribonucleotides, a critical reaction for viral DNA synthesis [12,13]. In this reaction, flavodoxin is proposed to mediate the two-electron reduction of the di-ferric cluster of RNR via two successive one-electron transfers. However, this has never been shown experimentally. 

Previously, we have reported that genes for multiple and unique monooxygenase cytochromes P450 enzymes are encoded in many genomes of giant viruses (*Megavirales*) including *Mimiviridae* and *Pandoraviridae* [14]. Therein, we also reported that P450 genes are encoded in other viral genomes including a herpesvirus (*Ranid herpesvirus 3*) and a *Mycobacterium* phage (*Mycobacterium* phage Adler). However, genes encoding known P450 redox partners (cytochrome P450 reductase or POR, ferredoxin and ferredoxin reductase, flavodoxin and flavodoxin reductase) were not found in any viral/phage genome thus far described, implying that host redox partners may drive viral P450 activities [14]. Herein, we now show that a bacteriophage 0305φ8-36 [15,16] encodes FMN-bound flavodoxin that can drive bacterial P450 activity, namely CYP170A1 oxidation, to produce the sequiterpene albaflavenone in the soil bacterium *Streptomyces coelicolor*. This result strongly suggests that phage flavodoxin can influence the biochemical activity of bacterial host enzymes, including P450 enzymes. The environmental relevance of bacteriophages for microbial systems has been widely explored in aquatic environments, but our current understanding of the role of phage(s) in soil ecosystems remains limited. Our results emphasize the importance of accounting for the effect of phage-encoded enzymes on interacting with and manipulating host metabolic pathways and therefore community function, with a novel and exciting example here from the soil microbiome.

## 3. Materials and Methods

### 3.1. Bioinformatic and Phylogenetic Analysis 

Fld sequences were retrieved from the NCBI sequence databases and BLAST searches employed to identify candidate phage Fld genes. Inferred protein sequences were initially aligned using ClustalOmega [17] and scored with T-Coffee [18]. Alignments were then visualized in BioEdit [19]. Final sequence alignment was constructed as described previously [20]. Virus names are those in the original reports or Genbank. Note that the sequence identities in the text are pairwise amino acid identities based on the length of the shortest protein in the alignment. Identity values used in the tables are based on multiple sequence alignments and are usually not more than 1% different from values stated in the text. Phylogenetic analyses were conducted on a mixture sequences obtained from BLAST searches of the NCBI databases, host sequences, and reference flavodoxin sequences. Sequences were aligned to the PFAM flavodoxin hidden Markov model (FMN_red, PF03358.18) using Clustal-omega v1.2.2 [17]. Maximum likelihood analyses were carried out using RAxML-NG (v1.1.0) [21]. Twenty independent ML searches were carried out followed by 1000 bootstrap replicates. Bootstrap support was assigned using the transfer bootstrap expectation method [22] and visualized using Figtree.

### 3.2. Cloning, Heterologous Expression and Purification of 0305φ8-36 Flavodoxin

The 0305φ8-36 flavodoxin gene was generated by DNA2.0 (Eurofins) and gene integrity confirmed by DNA sequencing. A unique *Nde*I restriction site was generated prior to the ATG start codon and a *Hin*dIII site following the TGA stop codon to facilitate cloning into the *E. coli* expression plasmid, pET17b. Additionally, a six residue polyhistidine tag was engineered into the C-terminus of the protein to allow purification of the expressed protein by Ni^2+^-NTA affinity chromatography. 0305φ8-36 Fld was expressed and purified using similar conditions described for P450 haemoproteins in our laboratory [23,24]. Briefly, transformed BL21(DE3)pLys cells were cultured overnight in Luria Bertani broth containing 100 μg/mL ampicillin. After inoculation (1:100) in 3 litres of Terrific Broth containing 100 μg/mL ampicillin, growth was carried out at 37 °C and 250 rpm for 6 h. Following the addition of riboflavin (1 mM final concentration) for flavin synthesis, flavodoxin expression was induced by the addition of isopropyl-β-D-thiogalactopyranoside (1 mM final concentration). Cell growth was continued for an additional 24 h at 27 °C and 190 rpm. Cells were harvested by centrifugation, resuspended in 250 mM sucrose, 50 mM Tris-HCl, pH 7.4 and disrupted by sonication. The insoluble cell debris was removed by centrifugation at 12,000× *g* and the supernatant applied to a 5 mL nickel affinity chromatography column (Qiagen) pre-equilibrated with 50 mM Tris-HCl, 100 mM NaCl, 20 mM imidazole, pH 7.4. Unbound protein was removed with further washes, and the His-tagged protein eluted in a single step with 50 mM Tris-HCl, 100 mM NaCl, 500 mM imidazole, pH 7.4. The eluted protein was concentrated to 2 mL using a 3 kDa molecular weight cut off Amicon centrifugal concentrator (Amicon, Merck) and glycerol (20% (*v*/*v*)) was added as a cryoprotectant. The purified protein was frozen at −80 °C until used.

### 3.3. Reconstitution of Streptomyces coelicolor CYP170A1 Activity with 0305φ8-36 Flavodoxin

*S. coelicolor* CYP170A1 (1 nmol), 0305φ8-36 flavodoxin or *E. coli* flavodoxin (10 nmol), and *E. coli* flavodoxin reductase (2 nmol) were reconstituted in 400 μL of 50 mM Tris-HCl buffer (pH 8.2) containing 20% (*v*/*v*) glycerol and epiisozizaene (20 nmol) plus 1% (*v*/*v*) Me_2_SO. Following incubation of this mixture for 5 min on ice, the reconstituted enzyme solution was placed in a shaking bath at 35 °C. The reaction was started by the addition of NADPH to a final concentration of 1 mM and 60 μL of an NADPH-regenerating system (glucose-6-phosphate (60 mM), β-nicotinamide adenine dinucleotide phosphate (3 mM), and glucose-6-phosphate dehydrogenase (0.2 unit)). The reaction was carried out for 1.5 h in a 10-mL test tube, at which time it was quenched with 4 μL of concentrated HCl and extracted three times with 400 μL of pentane:methylene chloride (4:1). The extracts were concentrated under a stream of N_2_, and 2 μL of extract was analyzed by gas chromatography/mass spectrometry (GC/MS). The mixtures were extracted and analyzed by GC/MS. Turnover rates for the epi-isozizaene oxidation was determined from triplicate experiments.

### 3.4. General Methods

The flavin content of Flds were determined by UV-visible and fluorescence spectrometry [25]. *S. coelicolor* CYP170A1 was expressed and purified as reported earlier [26]. To improve protein expression, CYP170A1 was coexpressed with molecular chaperones GroES/GroEL. Reduced carbon monoxide (CO) difference spectra for quantification of CYP170A1 content were measured and calculated according to the method described by Omura and Sato [27]. Additionally, recombinant *E.coli* flavodoxin and flavodoxin reductase were expressed and purified as described previously [28]. Protein quantification was performed by using the bicinchoninic acid assay. Unless otherwise stated, all chemicals were supplied by Sigma Chemical Company (Poole, Dorset, United Kingdom). UV-visible absorption spectra of purified flavodoxins and cytochrome P450 were recorded using a Shimadzu UV-2401 scanning spectrophotometer as described previously [26,29].

## 4. Results 

### 4.1. Bioinformatic and Phylogenetic Analysis of 0305φ8-36 ORF223 

Our earlier work has shown that some giant viruses and a *Mycobacterium* phage encode cytochrome P450 monooxygenase enzymes, however the associated genes for P450 electron transfer proteins have remained elusive [14]. Hence, we have looked in more detail for the presence of P450 redox partner genes in the virosphere. A BLASTP bioinformatic search of all published viral genomes, using flavodoxin amino acid sequences from a variety of bacterial sources, revealed the presence of a putative flavodoxin gene (ORF223) present in the genome of the bacteriophage 0305φ8-36. 0305φ8-36 has a genome size of 218 kb and contains 247 putative ORFs [15,16]. 0305φ8-36 infects the bacterium *Bacillus thuringiensis*, a soil bacterium commonly used as a biological pesticide [30]. The complete genome sequence of 0305φ8-36 was published in 2007 (Genbank accession no. EF583821). Examination of the genomic context of 0305φ8-36 ORF223 reveals that this putative flavodoxin gene is directly downstream of 2 ORFs predicted to encode α and β ribonucleotide reductase (ORFs 224 and 225 respectively) and that all three genes are translated from the same mRNA transcript (Figure 1), suggesting gene products that work in a complex.

The 0305φ8-36 ORF223 is predicted to encode a 147 amino acid polypeptide with a predicted molecular weight 16.5 kDa. The isoelectric point of 0305φ8-36 ORF223 was calculated to be approximately 4.9, resulting in an acidic protein similar to other flavodoxin proteins [1]. Amino acid sequence alignment revealed that key bacterial flavodoxin residues essential to binding and stabilising of the FMN cofactor, as determined by protein crystallography analysis of *E. coli* FldA, are also conserved in the 0305φ8-36 phage flavodoxin protein. These include Thr14 and Thr17 (Thr12 and Thr15 in *E. coli* flavodoxin), whose side chains form hydrogen bonds to stabilise the FMN phosphate group. Also conserved were phage Trp61 and Tyr97 (Trp57 and Tyr94 in *E. coli* flavodoxin), which form tight coplanar interactions with the isoalloxazine ring of FMN (Figure 2).

BLASTP analysis of 0305φ8-36 ORF223 against the public databases resulted in the best matches (46–37% sequence identity) to predicted flavodoxin proteins from soil dwelling bacteria including *Bacillus massiliosenegalensis* (46.5% identity; E-value 2 × 10^−35^); *Bacillus alkalicellulosilyticus* (43.2% identity; E-value 5 × 10^−33^); *Rhodococcus qingshengii* (43.7% identity; E-value 2 × 10^−32^) and *Mycobacteroides abscessus subsp. Abscessus* (42.3% identity; E-value 5 × 10^−29^). BLASTP analysis of 0305φ8-36 ORF223 against *B. thuringiensis* genomes revealed matches with a number of predicted flavodoxin proteins but with lower sequence identity (34–31% sequence identity; 95–65% sequence coverage e.g., *Bacillus thuringiensis* MC28, 33% identity, E-value 7 × 10^−16^). The closest homologues of most of the 0305φ8-36 virion protein-encoding genes and a few replicative genes were found to reside in a single segment of the chromosome of *B. thuringiensis* serovar *Israelensis*. Sequence identity between *B. thuringiensis* serovar *Israelensis* flavodoxin and 0305φ8-36 flavodoxin is only 31%. Thus, it is difficult to say with any great certainty whether the 0305φ8-36 phage captured ORF223 by horizontal gene transfer from the host *B. thurigiensis* or from another soil bacterial species. 

We also undertook a BLASTP analysis to search for the presence of flavodoxin genes in other viruses. Putative flavodoxin genes were identified in the following phages: *Psychrobacillus* phage Perkons (34.7% identity; E-value 2 × 10^−9^); *Enterococcus* phage EF1 (31.3% identity; E-value 3 × 10^−8^); *Bacillus* phage vB_BcoS-136 (32% identity; E-value 3 × 10^−7^) and *Staphylococcus* phage vB_SepS_SEP9 (28.2% identity; E-value 8 × 10^−4^) and many identical or near identical isolates in Listeria phage (19 isolates; see also Table 1 for a list of phage genomes with encoded flavodoxin genes). We performed maximum likelihood (ML) phylogenetic analyses of 0305phi8-36 along with these additional phage proteins and the respective hosts of most of the phages with flavodoxins. Most of the phage proteins cluster together, in many cases with the host from which they were isolated (e.g., *Listeria*, *Psychrobacillus*). In other cases, notably 0305phi8-36, and *Staphylococcus*, *Enterococcus*, and *Arcanobacterium* phages, the phages are not clustered with their hosts, perhaps reflecting an origin in other soil bacteria (Figure 3).

### 4.2. Expression, Purification and Spectral Analysis of Recombinant 0305φ8-36 ORF223

The heterologous expression of histidine-tagged 0305φ8-36 ORF223 in *E. coli* and purification using Ni^2+^-NTA affinity chromatography yielded an average of 10 mg of purified flavodoxin protein per liter of culture. Cell fractionation by ultracentrifugation at 100,000× *g* revealed 0305φ8-36 ORF223 to be a soluble protein being located in the cytosolic fraction. No FMN-bound protein was detected in the isolated *E. coli* membranes. After two consecutive Ni^2+^-affinity chromatography purification steps a homogeneous band was observed on sodium dodecyl sulphate-polyacrylamide gel electrophoresis at approximately ~20 kDa, compared with the predicted molecular mass of the protein (16.5 kDa). This is probably due to inclusion of the engineered His tag into the recombinant protein, which results in mobility retardation in the PAGE gel. The UV-visible absorption spectrum of purified 0305φ8-36 ORF223 showed maxima at 360 and 460 nm with a shoulder at 490 nm typical of oxidized Flds (Figure 4). The spectra generated for 0305φ8-36 ORF223 protein was similar with those produced for control purified *E. coli* flavodoxin (Figure 4; [23,26]) and these characteristics are typical of other purified and spectrally characterized flavodoxin proteins described in the literature [11,28].

### 4.3. Characterization of the Catalytic Activity of 0305φ8-36 Flavodoxin in P450-Mediated Monooxygenation

In the soil bacterium *S. coelicolor*, cytochrome P450 170A1 (CYP170A1) catalyzes the sequential allylic oxidation of epi-isozizaene to the sequiterpene, albaflavenone (Figure 5A). Because the endogenous reductase complex is not known for CYP170A1, we previously used the *E. coli* flavodoxin and flavodoxin reductase system to drive this monooxygenase activity [26]. To investigate if 0305φ8-36 flavodoxin is functionally active we reconstituted *S. coelicolor* CYP170A1 enzymatic activity, replacing *E. coli* flavodoxin in the *E. coli* flavodoxin/flavodoxin system with the viral flavodoxin protein. Analysis of the CYP170A1-catalyzed reaction products by GC/MS revealed the time-dependent formation of two isomeric albaflavenol products with *m*/*z* 220 (*t*_R_ 11.8 min; *t*_R_ 12 min) and the ketone albaflaveone sequiterpene product with *m*/*z* 218 (*t*_R_ 12.7 min) (Figure 5B). Using the *E. coli* flavodoxin/flavodoxin reductase couple, the turnover number was calculated to be ~0.36 ± 0.04 min^−1^ for the conversion of epi-isozizaene to albaflavenone. When the *E. coli* flavodoxin was replaced with 0305φ8-36 flavodoxin the turnover number was ~0.29 ± 0.03 min^−1^. Negative control incubations lacking CYP170A1, flavodoxins, or NADPH did not generate any products.

## 5. Discussion

When first discovered, flavodoxin was characterized as a key electron transfer protein functioning in photosynthetic cyanobacteria growing under limiting iron conditions [2,3]. Since that time flavodoxins have been discovered in numerous non-photosynthetic organisms, mainly *Bacteria* and in *Archaea*, where they function in diverse metabolic pathways. For example, in *E.coli* flavodoxin is involved in the biosynthesis of methionine through activation of cobalamine-dependent methionine synthase [60]; in *B. subtilis* flavodoxin plays a role in electron transfer to drive the cytochrome P450 enzyme (CYP107H1) involved in the biosynthesis of the water-soluble vitamin biotin [61]; and in the diazotroph *Azotobacter vinelandii* flavodoxin transfers reducing equivalents to the nitrogenase enzyme involved in nitrate reduction [62]. In the archaeon *Methanosarcina acetivorans* a flavodoxin plays a role in electron transfer in the metabolic pathway converting acetate to methane [63]. In bacterial cytochrome P450 systems flavodoxin, while driving P450 as a stand-alone protein, can be structurally linked to P450 in three ways: (i) fused to a P450 domain e.g., *Rhodococcus rhodochrous* XplA which is used in biotechnology to degrade the military explosive hexahydro-1,3,5-trinitro-1,3,5-triazine [64] (ii) fused to both the P450 domain and a flavodoxin reductase domain e.g., *Rhodococcus ruber* CYP116B3 a protein which can metabolize an array of hydrocarbons including the environmental pollutant xylene [65] and (iii) genetically organised in an operon with P450 and flavodoxin reductase e.g., *Citrobacter braakii* CYP176A1 catalyzes the conversion of cineole to 6β-hydroxycineole in energy metabolism [66]. In all cases genetic organisation is thought to favour optimal flavodoxin electron transfer during P450 catalysis.

Microsynteny analysis of phage flavodoxin genes reveals that in all cases that we examined, the flavodoxin gene is linked to or in the close neighbourhood of genes encoding α and β ribonucleotide reductase (Figure 6). This strongly suggests that the endogenous function of phage flavodoxin is to provide reducing equivalents to these enzymes during the catalytic removal of the 2′-hydroxyl group of the ribose ring of nucleoside diphosphates [67]. This reaction occurs during the formation of deoxyribonucleotides from ribonucleotides, which are necessary for DNA synthesis during phage replication. 

0305φ8-36 is a bacteriophage that infects the soil bacterium *B. thuringiensis* [15,16]. Examining the metabolic pathways of this bacterium, our analysis of the sequenced genome of the *B. thuringiensis* reference strain Bt407 [68] revealed the presence of three cytochrome P450 genes: *CYP102A5*, *CYP106B4* and *CYP107J3*. Furthermore, other strains of *B. thuringiensis*, infected by phage encoding flavodoxin [31], have been shown to encode P450 family members: *CYP102* (*CYP102A8*); *CYP106* (*CYP106B1*, *B5*); *CYP107DY4*; *CYP107J* (*CYP107J2*); *CYP109T5* and *CYP2467A1* [69] (D. Nelson personal communication). It is feasible that the infected *B. thuringiensis* strains can have their P450 enzyme activities enhanced, modified or inhibited by the phage flavodoxin. The phenotypic results will be dependent on the host P450 activity. CYP102A enzymes are generally self-sufficient P450s with a fused C-terminal reductase domain that are known to metabolize fatty acids and lipids, although no endogenous function for a CYP102 has been determined [24]. Similarly, the endogenous functions of the CYP106 and CYP107 enzymes are unknown. However, individual recombinant CYP106 and CYP107 enzymes have been shown to selectively hydroxylate a variety of steroid molecules including testosterone, progesterone and their derivatives [70]. Phages that encode flavodoxin can also infect other *Bacillus* species including *B. cereus*, *B. subtilis* and *B. anthracis* (Figure 6; [71]. Previous in silico analysis of P450s in 128 Bacillus species revealed the presence of 507 P450s grouped into 13 P450 families and 28 subfamilies with no P450 family found to be conserved across *Bacillus* species [72]. Bioinformatic analysis of those *Bacillus* P450 genes revealed that a limited number of P450 families dominate - CYP102, CYP106, CYP107, CYP109, CYP134, CYP152, and CYP197. Conversely, no P450 genes have been found to be encoded in any sequenced genome of a *Listeria* species [69]. However, of the more than 90 *Staphylococcus* species that have been sequenced, nine have been shown to encode P450. *Staphylococcus* P450s are limited to members of the CYP134 and CYP152 families [73]. Of note is P450 P450 OleT_SA_, a cytochrome P450 enzyme from *Staphylococcus aureus* that catalyzes the oxidative decarboxylation and hydroxylation of fatty acids to generate terminal alkenes and fatty alcohols [74]. It is possible that *Staphylococcus* phage flavodoxin can interact with *S. aureus* OleT and impact this catalytic activity. 

Data are now revealing a remarkable diversity of metabolic genes in viruses, including many involved in sphingolipid biosynthesis, fermentation, and nitrogen metabolism [75,76]. For example, numerous giant viruses are predicted to encode the components of glycolysis and the tricarboxylic acid cycle, suggesting that they can re-program fundamental aspects of their host’s central carbon metabolism [77,78]. However, many of these viral enzymes also require cofactors to function. For example, aconitase and fumarase require Fe-S clusters—and other enzymes show other prosthetic group requirements (FAD/FMN/heme). This has led to a re-evaluation of the complex metabolic capabilities of viruses, particularly given their roles as important drivers of global bio-geochemical cycles [79]. 

In this study, we have found that a phage encoded FMN-containing flavodoxin can drive bacterial P450 activity. Similar phages infect different species of soil-inhabiting bacteria, mainly of the *Bacillus* genus and this suggests that the impacts of this metabolic activity may be widespread. Our results clearly show that viruses can engage with and possibly hijack host oxidative enzyme capacity. Historically, viruses were considered as metabolically inactive, non-living entities in comparison to the metabolic complexity displayed by cellular life. Their phenotypic impact was thought limited to merely affecting host mortality. Recent work has cast this aspersion aside, as we have come to realize that close metabolic integration appears to be required for viruses to subdue and coerce their unwitting hosts. This work clearly demonstrates the potential for viruses to not just manipulate ‘virus centric’ replication activities, but broader host metabolic activities and cellular function(s) as well. The large number of viral metabolic genes, including the different classes of redox proteins, requires further research into how these virus-specific enzymes affect both host physiology and the global microbiome and biogeochemical cycling. 

## Figures and Tables

**Figure 1 biomolecules-12-01107-f001:**
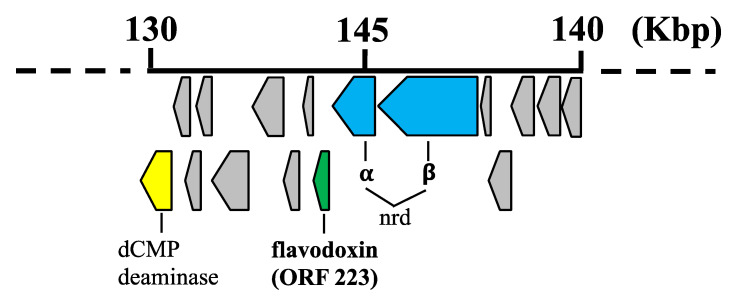
Schematic diagram showing the genomic region of bacteriophage 0305φ8-36 where the flavodoxin gene is located. The scale is in kilobase pairs. Each arrow represents an individual orf. The flavodoxin gene, colored green, is 0305φ8-36 ORF number 223. ORFs colored yellow and blue have been tentatively assigned dCMP deaminase and α and β-ribonucleotide reductase (nrd), respectively. Arrows shown in grey represent genes predicted to encode virion proteins of unknown function.

**Figure 2 biomolecules-12-01107-f002:**
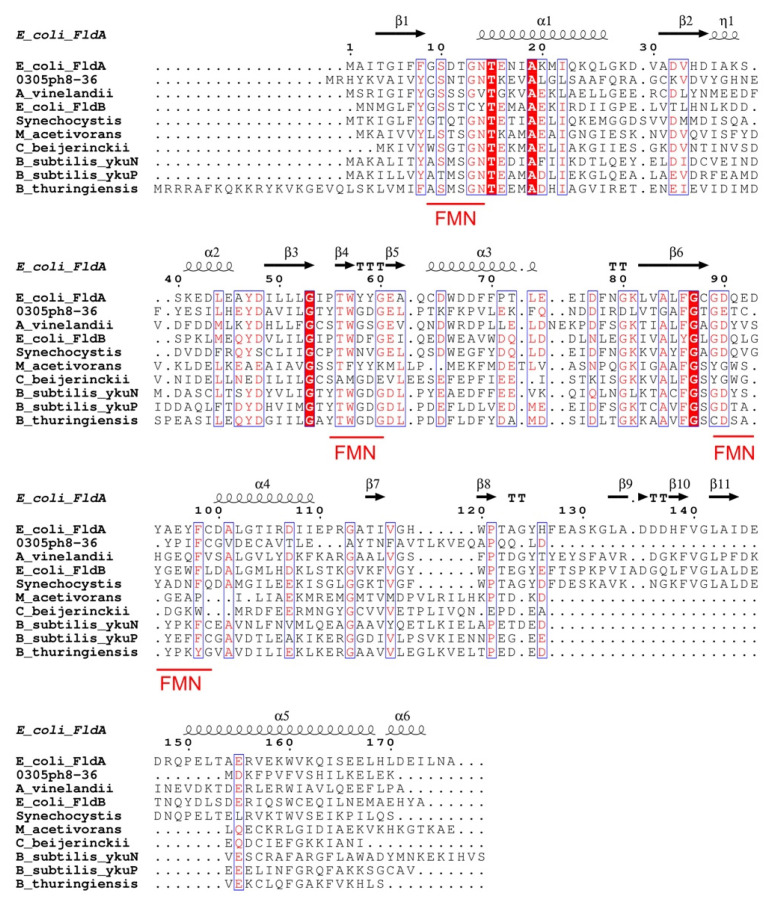
Amino acid sequence alignment of 0305φ8-36 phage flavodoxin with selected cellular flavodoxins. Alignment was generated using the Clustal Omega program via the EBI (https://www.ebi.ac.uk/Tools/msa/clustalo/, accessed on 22 July 2022). The flavodoxin sequences used were: long chain flavodoxins, *E. coli* (strain K12) FldA (accession number: P61949), *E. coli* (strain K12) FplB (P0ABY4) and short chain flavodoxins, *B. subtilis* (strain 168) ykuN (O34737), *B. subtilis* ykuP (strain 168) (O34589), *Methanosarcina acetivorans* (strain ATCC 35395) (Q8TPV5), *B. thuringiensis* serovar *israelensis* (strain ATCC 35646) (Q3ESQ3), *Azotobacter vinelandii* (strain DJ/ATCC BAA-1303) (P52964), *Clostridium beijerinckii* (Clostridium MP) (P00322) and *Synechocystis* sp. (strain PCC 6803/Kazusa) (P27319). Residues indicated by asterisks are conserved in all sequences. Residues indicated by single and double dots are of similar and highly similar chemical character, respectively, according to criteria set in the program. The residues comprising the three FMN-binding loops are underlined in red.

**Figure 3 biomolecules-12-01107-f003:**
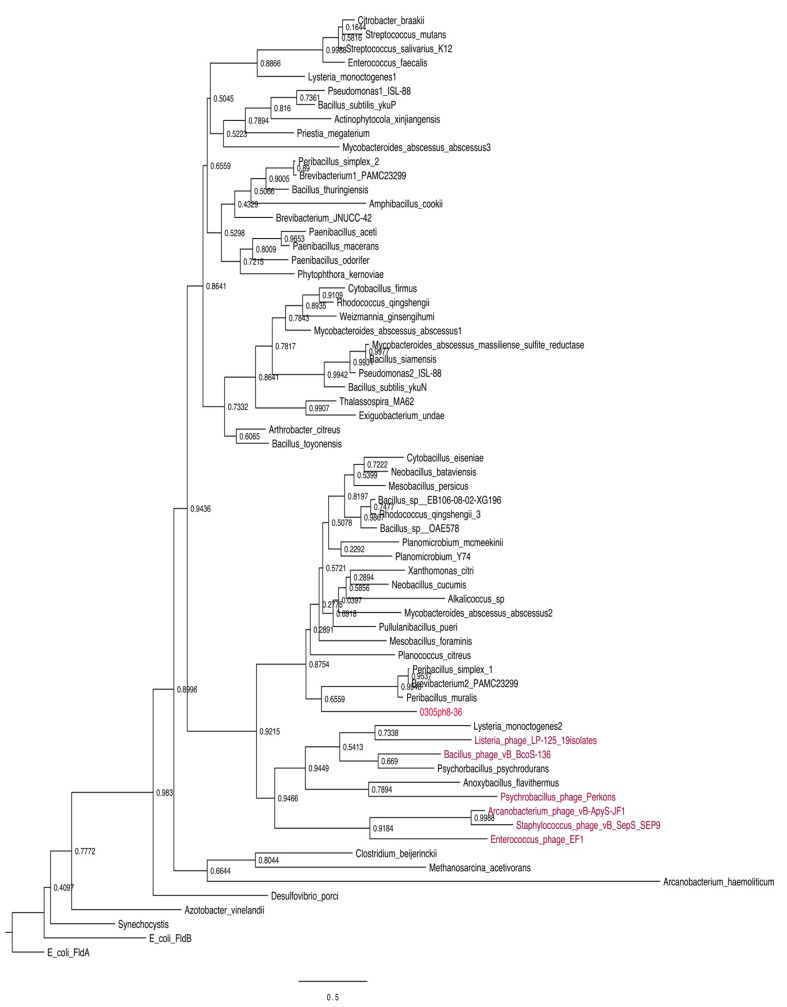
Relationship of 0305φ8-36 flavodoxin with other phage and bacterial host flavodoxins. Phylogenetic tree of 0305φ8-36 flavodoxin with additional phage flavodoxins and the respective hosts of most of the phages with flavodoxins. *Numbers* indicate the bootstrap probability values for the branch topology shown.

**Figure 4 biomolecules-12-01107-f004:**
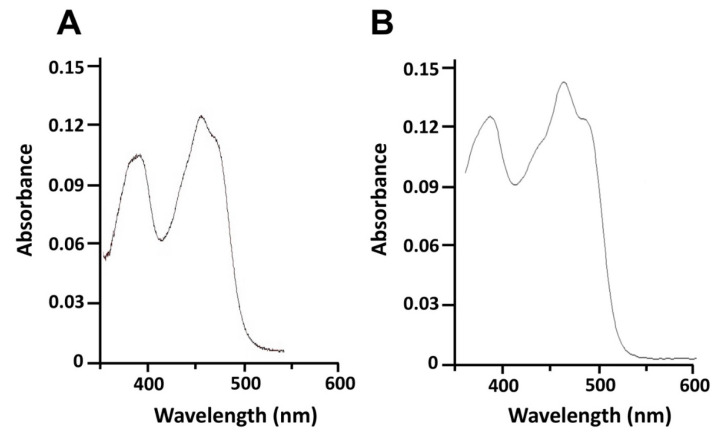
UV-visible absorbance spectra of purified *E. coli* and 0305φ8-36 phage flavodoxins. (**A**) Oxidized spectrum obtained for purified *E. coli* flavodoxin (15 μM). (**B**) Oxidized spectrum obtained for purified phage 0305φ8-36 flavodoxin (15 μM).

**Figure 5 biomolecules-12-01107-f005:**
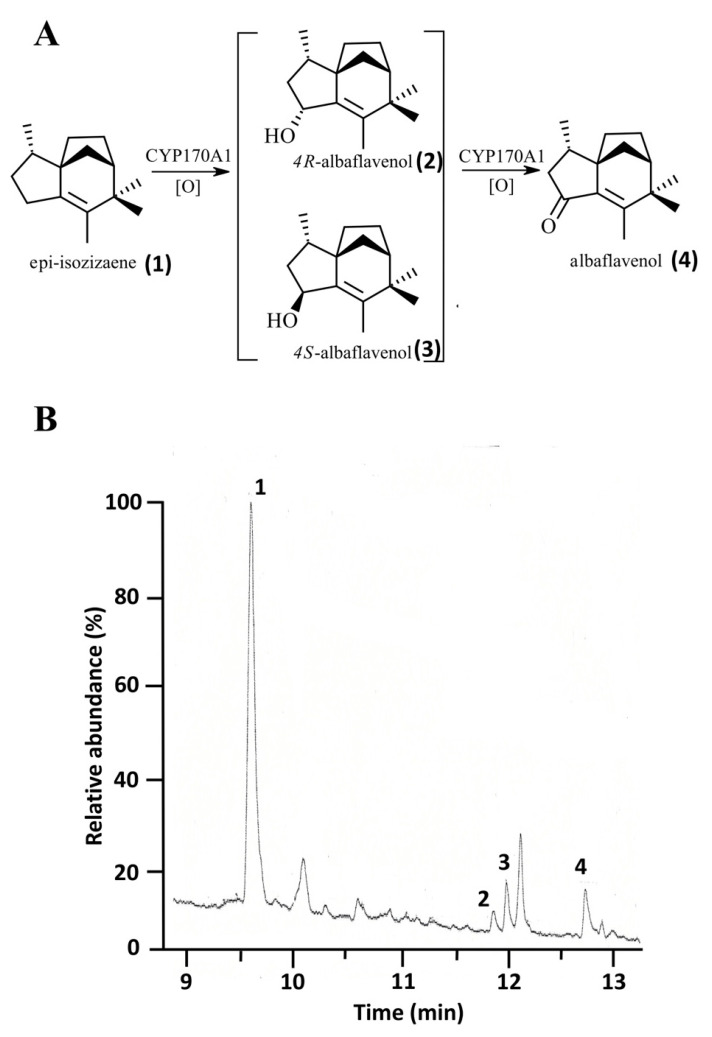
Epi-isozizaene allylic oxidation by CYP170A1 supported by 0305φ8-36 flavodoxin. (**A**) Metabolic pathway for the formation of albaflavenone catalysed by CYP170A1. (**B**) In vitro catalytic activity of CYP170A1 supported by 0305φ8-36 and *E. coli* flavodoxin reductase. Product profile was obtained using individual component concentrations as follows: 1 nmol CYP170A1, 10 nmol 0305φ8-36 flavodoxin or 10 nmol *E. coli* flavodoxin, 2.0 nmol *E. coli* flavodoxin reductase and 40 nmol epi-isozizaene. Three products are noted, **2** and **3** (4(R)-albaflavenol and 4(S)-albaflavenol epimers respectively) and **4** (albaflavenone), respectively; substrate epi-isozizaene as **1**. Experiments were performed in triplicate. In negative control incubations (without CYP170A1 or without flavodoxin/flavodoxin reductase), no product formation was observed (data not shown).

**Figure 6 biomolecules-12-01107-f006:**
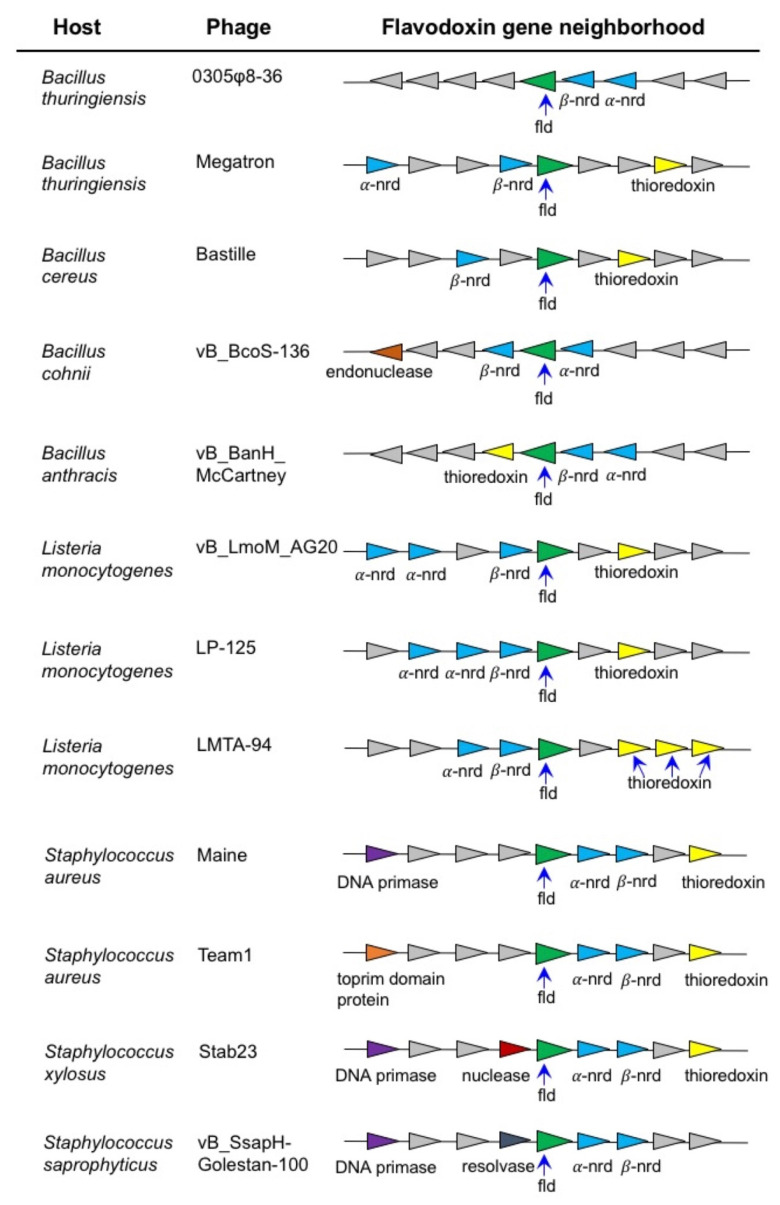
Microsynteny around phage flavodoxin genes. Comparison of the genetic *loci* and surrounding genes for flavodoxins in *Bacillus* phages, *Listeria* phages and *Staphylococcus* phages. Each phage flavodoxin gene (fld) is colored green and α and β ribonucleotide reductase genes (α and β nrb) are colored blue. In each *Bacillus, Listeria* and *Staphylococcus* phage genomes examined the fld gene was next to or close to α and β nrb genes, strongly suggesting functional linkage.

**Table 1 biomolecules-12-01107-t001:** Flavodoxin genes found in *Bacillus*, *Listeria* and *Staphylococcus* phage genomes.

Class	Phage	Phage Host	Sample Location	Predicted Phage ORFs	Flavodoxin aa	Genbank Identifier	Ref.
** *Bacillus* **	0305φ8-36	*Bacillus thuringiensis*	soil	246	148	ABS83781.1	[16]
	Hakuna	*Bacillus thuringiensis*	soil	294	159	YP_009036585.1	[31]
	Megatron	*Bacillus thuringiensis*	soil	290	159	YP_009036208.1	[31]
	Riley	*Bacillus thuringiensis*	soil	290	149	YP_009055891.1	[31]
	CAM003	*Bacillus thuringiensis*	soil	287	150	YP_009037037.1	[31]
	Evoli	*Bacillus thuringiensis*	soil	294	150	YP_009035659.1	[31]
	Hoody T	*Bacillus thuringiensis*	soil	270	153	YP_009035330.1	[31]
	Troll	*Bacillus thuringiensis*	soil	289	149	YP_008430917.1	[31]
	SalinJah	*Bacillus cereus*	soil	292	149	ANH50605.1	[32]
	Eyuki	*Bacillus cereus*	soil	300	159	YP_009212082.1	[33]
	AvesoBmore	*Bacillus thuringiensis*	soil	301	149	YP_009206486.1	[33]
	BigBertha	*Bacillus thuringiensis*	soil	291	153	YP_008771155.1	[34]
	Spock	*Bacillus thuringiensis*	soil	283	149	YP_008770352.1	[35]
	Belinda	*Bacillus thuringiensis*	soil	295	159	ANM46069.1	[36]
	Tomato	*Bacillus thuringiensis*	soil	200	159	QLF85935.1	[37]
	AaronPhadgers	*Bacillus thuringiensis*	soil	301	159	ASR78796.1	[38]
	Beyonphe	*Bacillus cereus*	water	300	149	QDH49824.1	[38]
	Bubs	*Bacillus thuringiensis*	soil	302	159	ASR78601.1	[38]
	Kamfam	*Bacillus thuringiensis*	soil	293	150	AXQ67322.1	[38]
	PIPsBest	*Bacillus thuringiensis*	soil	301	159	ASR78379.1	[38]
	ALPS	*Bacillus thuringiensis*	soil	295	159	QDH50122.1	[38]
	Phireball	*Bacillus thuringiensis*	soil	299	159	QDH49415.1	[38]
	OmnioDeoPrimus	*Bacillus thuringiensis*	soil	299	159	AXQ67477.1	[38]
	Zainny	*Bacillus thuringiensis*	soil	303	159	ASR79375.1	[38]
	vB_BthM-Goe5	*Bacillus thuringiensis*	soil	272	150	AZF89241.1	[39]
	SageFayge	*Bacillus thuringiensis*	soil	300	159	YP_009280944.1	[40]
	Nemo	*Bacillus thuringiensis*	soil	301	159	YP_009287019.1	[40]
	Nigalana	*Bacillus thuringiensis*	soil	302	159	YP_009282535.1	[40]
	DIGNKC	*Bacillus thuringiensis*	soil	291	159	AMW62869.1	[40]
	Zuko	*Bacillus thuringiensis*	soil	294	159	AMW62552.1	[40]
	Phrodo	*Bacillus thuringiensis*	soil	288	149	YP_009290006.1	[40]
	NotTheCreek	*Bacillus thuringiensis*	soil	296	159	YP_009284467.1	[40]
	Juglone	*Bacillus thuringiensis*	soil	293	149	AMW61744.1	[40]
	Vinny	*Bacillus thuringiensis*	soil	297	150	AMW61891.1	[40]
	DirtyBetty	*Bacillus thuringiensis*	soil	302	159	YP_009285086.1	[41]
	Kida	*Bacillus thuringiensis*	soil	304	159	YP_009279309.1	[41]
	TsarBomba	*Bacillus thuringiensis*	soil	247	151	YP_009206942.1	[42]
	HonestAbe	*Bacillus thuringiensis*	soil	286	159	AUV57777.1	[43]
	Taffo16	*Bacillus thuringiensis*	soil	284	149	ASZ75860.1	[43]
	Anthony	*Bacillus thuringiensis*	soil	279	150	ASU00988.1	[43]
	Saddex	*Bacillus thuringiensis*	soil	208	159	AXF41931.1	[44]
	Janet	*Bacillus thuringiensis*	soil	285	150	ASR79938.1	[45]
	OTooleKemple52	*Bacillus thuringiensis*	soil	291	153	ASR79614.1	[45]
	SBP8a	*Bacillus cereus*	soil	298	146	AOZ62389.1	DS *
	QCM8	*Bacillus cereus*	soil	288	151	AOZ62044.1	DS *
	BJ4	*Bacillus cereus*	soil	298	146	AOZ61763.1	DS *
	Bastille	*Bacillus cereus*	soil	273	150	YP_006907364.1	DS *
	Chotacabras	*Bacillus thuringiensis*	soil	285	149	QEM43180.1	DS *
	vB_BcoS-136	*Bacillus cohnii*	lake sediment	238	146	AYP68319.1	DS *
	Flapjack	*Bacillus thuringiensis*	soil	288	149	ARQ95040.1	DS *
	vB_BanH_ RonSwanson	*Bacillus anthracis*	soil	-	202	UGO50429.1	DS *
	vB_BanH_ Emiliahah	*Bacillus anthracis*	soil	-	174	UGO49205.1	DS *
	vB_BanH_JarJar	*Bacillus anthracis*	soil	-	202	UGO48939.1	DS *
	vB_BanH_ McCartney	*Bacillus anthracis*	soil	-	174	UGO47691.1	DS *
	vB_BanH_ Abinadi	*Bacillus anthracis*	soil	-	159	UGO46389.1	DS *
	Anthos	*Bacillus thuringiensis*	soil	290	149	QPY77365.1	DS *
	Smudge	*Bacillus thuringiensis*	soil	292	159	ANI24755.1	DS *
	BC-T25	*Bacillus* sp.	soil	236	151	QEG04194.1	DS *
** *Listeria* **	vB_LmoM_AG20	*Listeria monocytogenes*	abbatoir	178	141	YP_007676786.1	DS *
	LP-124	*Listeria monocytogenes*	silage	188	150	YP_009784573.1	[46]
	LP-064	*Listeria monocytogenes*	silage	188	150	YP_009592673.1	[46]
	LP-083-2	*Listeria monocytogenes*	silage	189	150	YP_009044596.1	[46]
	LP-048	*Listeria monocytogenes*	silage	177	150	YP_009042933.1	[46]
	LP-125	*Listeria monocytogenes*	silage	189	150	YP_008240106.1	[46]
	LP-Mix_6.2	*Listeria monocytogenes*	laboratory	197	150	QNL32088.1	[47]
	LP-Mix_6.1	*Listeria monocytogenes*	laboratory	195	150	QNL31890.1	[47]
	LP-039	*Listeria monocytogenes*	silage	201	150	QEP53123.2	[48]
	LP-066	*Listeria monocytogenes*	silage	189	150	QDK04972.2	[48]
	LMTA-94	*Listeria monocytogenes*	laboratory	189	150	AID17150.1	DS
** *Staphylococcus* **	Maine	*Staphylococcus aureus*	pig barn swab	219	130	QEM41386.1	[49]
	MarsHill	*Staphylococcus aureus*	pig barn swab	262	134	QQM14610.1	[50]
	Madawaska	*Staphylococcus aureus*	pig barn swab	264	134	QQO92731.1	[50]
	Machias	*Staphylococcus aureus*	pig barn swab	263	140	QQO92468.1	[50]
	phiIPLA-RODI	*Staphylococcus aureus*	sewage	213	139	AJA42083.1	[51]
	phiIPLA-C1C	*Staphylococcus aureus*	sewage	203	132	YP_009214514.1	[51]
	Team1	*Staphylococcus aureus*	hospital	217	143	YP_009098273.1	[52]
	MCE-2014	*Staphylococcus aureus*	sewage	204	130	P_009098058.1	[53]
	GH15	*Staphylococcus aureus*	sewage	214	130	YP_007002257.1	[54]
	Metroid	*Staphylococcus* sp.	soil	254	143	KE56191.1	[55]
	pSco-10	*Staphylococcus cohnii*	duck feces	131	130	ANH50479.1	[56]
	BT3	*Staphylococcus aureus*	animal	232	139	QVD58098.1	[57]
	Stab23	*Staphylococcus xylosus*	sewage	247	130	VEV88569.1	[58]
	vB_Sau_S24	*Staphylococcus aureus*	soil	209	130	ARM69410.1	[59]
	vB_Sau_Clo6	*Staphylococcus aureus*	sewage	213	130	ARM69197.1	[59]
	vB_SsapH-Golestan-100	*Staphylococcus saprophyticus*	-	192	133	BDA81541.1	DS *
	phiSA_BS1	*Staphylococcus* sp.	dairy farm	200	129	YP_009799552.1	DS *
	vB_SsapH-Golestan-105-M	*Staphylococcus saprophyticus*	-	203	149	BDA82285.1	DS *
	SA3	*Staphylococcus aureus*	sewage	223	130	ASZ78055.1	DS *
	PALS_1	*Staphylococcus aureus*	animal	191	130	QDJ97591.1	DS *

* DS direct submission to Genbank.

## Data Availability

The data used to support the findings of this research are available upon request.

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
