# Peer review of "Characterization of a Virally Encoded Flavodoxin That Can Drive Bacterial Cytochrome P450 Monooxygenase Activity"

_biomolecules, 2022, doi:10.3390/biom12081107_

Round 1

Reviewer 1 Report

see the attached file.

Reviewer 2 Report

This is an excellent article with meticulous work. I have no hesitation to recommend the article for publication in the current format. 

I congratulate the authors for their excellent work.

Regards,

Syed

Reviewer 3 Report

The authors declare that this study aims to probe the hypothesis that viral flavodoxins can support the activity of cytochrome P450 monooxygenases of the host bacteria. Indeed, one can imagine a situation where the interaction of bacterial flavodoxins with host cytochromes P450 can lead to unusual physiological effects. However, the selection of the donor-acceptor pair for this study raises significant concerns. CYP170A1 has no close homologue in Bacillus thuringiensis, and the pair of 0305φ8-36 flavodoxin with CYP170A1 is entirely non-physiological. On the other hand, any flavodoxin would likely be capable of supporting the activity of any P450 to some extent. Furthermore, the turnover number for the pair of CYP170A1 with viral flavodoxin reported in the manuscript (0.29 min-1) is quite low. In the absence of comparison with the turnover of the enzyme in physiological conditions, these results do not provide any evidence that pairing bacterial P450 with viral flavodoxin may be physiologically important. So, probing the pair of CYP170A1 with 0305φ8-36 flavodoxin does not add much to the science of phage-host interactions and does not support the author’s hypothesis. 

A much better explanation of the setup of this study and a deeper analysis of possible viral flavodoxin-bacterial P450 pairs in Bacillus thuringiensis (the host bacteria for 0305φ8-36 phage) and Streptomyces coelicolor (the host of CYP170A1) is needed to prove the scientific significance of this study.  For example, are there any flavodoxins in viruses infecting S. coelicolor? If there are, and these flavodoxins have close homology with 0305φ8-36 protein, then, might be, the reader may better appreciate this study.

On the other hand, the design of this study would look more cogent if the authors could better analyze possible physiological consequences of interactions of viral flavodoxin with P450 proteins of Bacillus thuringiensis. An attempt of this analysis found on page 16 (lines 323 - 353) does not sound convincing, especially in view that the authors forgot to mention that CYP102A5, which possible interactions with viral flavodoxins they discuss, is a self-sufficient P450 enzyme. It contains a flavoprotein domain and thus does not require any flavodoxin partners (see http://mokslozurnalai.lmaleidykla.lt/publ/1392-0146/2008/3/187-191.pdf). Therefore, possible interactions of CYP102A5 with viral flavodoxins seem physiologically irrelevant. 

Summarizing: better explanation of the design of this study and a deeper discussion of its results are needed to reveal the possible scientific value of this manuscript. 

Minor points:

Fig. 4: The difference in the amplitudes of the spectra of two flavodoxins shown in this figure looks puzzling. If the absorbance of the E.coli flavodoxin is consistent with its known extinction coefficient (~0.0842 µM-1cm-1 at 464 nm), the absorbance of the viral flavodoxin is considerably higher. Is this difference real? How did the authors determine the concentration of the viral protein? Another issue with this figure is why do the authors use different scaling at the X-axis for the left and right panels? May both spectra be presented in one panel? 

Lines 308-309: “In all cases, genetic organisation is thought to favour optimal flavodoxin electron transfer during P450 catalysis.” - I do not understand what the authors wanted to say in this sentence. What do “genetic organisation” mean, and how can it favour electron transfer? Please, explain.

Line 329: “CYP2467A1” - To my knowledge, there is no such a protein. It must be a typo.

Reviewer 4 Report

The authors present the firs characterization of a virally encoded flavodoxin. I general, the manuscript will be pf interest to those working with bacterial phages and with flavodoxin enzyme. However, there are some points to consider.

1.- Table 1 is not cited in the text and is not clear the intention to include it. 

2.- Which is the correct flavodoxin's ORF number, 223 (line 175) or 217 (line 183)?

3.- It should be desirable to include an image of the SDS-PAGE from the enzyme purification process.

4.- Storing the enzyme at -80°C, did not provoke enzyme unfolding? or if a cryoprotectant agent was used please indicate this in methodology.

5.- Did all the enzymes in the reaction mix were not damage after 1.5 hour at 35°C? 

6.- In the same context, what about the activity values obtained in similar systems? 

7.- In general, the kinetic evidence provided is limited, it is necessary to include the graphics from which the kinetic data were obtained (turnover).

Round 2

Reviewer 3 Report

Unfortunately, the authors did not address my primary concern in responding to my criticism. I'm pretty skeptical about any rationale for probing activity in an entirely non-physiological pair of  0305φ8-36 flavodoxin with CYP170A1. First, these two proteins never meet in nature. Second, it is highly likely that any P450 can be reduced with some efficiency by any flavodoxin, and thus any P450-flavodoxin pair will exhibit some enzymatic activity. In my opinion, the idea of this study is quite far-fetched, and its scientific value is problematic. However, given the large volume and high quality of the experimental material, I agree with other reviewers that it may be published.